# Transaction Entropy: An Alternative Metric of Market Performance

**DOI:** 10.3390/e25081140

**Published:** 2023-07-30

**Authors:** Hua Zhong, Xiaohao Liang, Yougui Wang

**Affiliations:** School of Systems Science, Beijing Normal University, Beijing 100875, China; 201931250002@mail.bnu.edu.cn (H.Z.); 202221250004@mail.bnu.edu.cn (X.L.)

**Keywords:** market uncertainty, transaction entropy, market performance, price filtering mechanism, willingness price

## Abstract

Market uncertainty has a significant impact on market performance. Previous studies have dedicated much effort towards investigations into market uncertainty related to information asymmetry and risk. However, they have neglected the uncertainty inherent in market transactions, which is also an important aspect of market performance, besides the quantity of transactions and market efficiency. In this paper, we put forward a concept of transaction entropy to measure market uncertainty and see how it changes with price. Transaction entropy is defined as the ratio of the total information entropy of all traders to the quantity of transactions, reflecting the level of uncertainty in making successful transactions. Based on the computational and simulated results, our main finding is that transaction entropy is the lowest at equilibrium, it will decrease in a shortage market, and increase in a surplus market. Additionally, we make a comparison of the total entropy of the centralized market with that of the decentralized market, revealing that the price-filtering mechanism could effectively reduce market uncertainty. Overall, the introduction of transaction entropy enriches our understanding of market uncertainty and facilitates a more comprehensive assessment of market performance.

## 1. Introduction

The concept of the market holds great significance in economics and serves as a fundamental basis for research of economics [1]. Understanding the mechanisms by which markets function has profound implications for decision making, policy formulation, and economic development. The research on market primary functioning has long been focused on two key aspects: price formation and market efficiency.

In the context of market price, this is determined by the interaction between sellers and buyers. In a perfect competitive market, the market price is deemed to be at the cross point of the supply and demand curves. Therefore, the factors that influence these curves, such as the willingness of the market participants and information dissemination, have impacts on the level of the market price. Regarding market efficiency, market surplus is usually used to measure it. Market surplus represents the total welfare generated by transactions between sellers and buyers. An increase in market surplus signifies an improved efficiency in market transactions and a more optimal allocation of resources.

However, most analyses of price formation and market efficiency are typically conducted under the assumption of ideal conditions, without accounting for the uncertainties faced by the participants in a market. Several studies have verified the existence of uncertainties in market transactions [2,3]. During an actual transaction process, each participant has limited access to information and cannot obtain complete knowledge about the counterparty’s information or the overall market situation. This inherent imperfection in information significantly influences the decision making and behavior of both parties involved, thereby increasing transactional uncertainty, which subsequently impacts market prices and market surplus. Therefore, understanding the mechanism of price formation and improving market efficiency have become the cornerstones of economic analyses, and we contend that incorporating uncertainty into a market can not only provide a more accurate description of market performance, but also enrich our understanding of market functioning.

Financial markets serve as the primary focus of market uncertainty analyses. In financial markets, the market participants make investment decisions based on expectations, which inherently carry a certain level of uncertainty. Therefore, uncertainty is a common and essential aspect of financial markets. These uncertainties, in turn, exert negative effects on market information efficiency [4]. To enhance market efficiency, it is crucial to understand and measure financial market uncertainty [5]. Information entropy is commonly used to measure such a kind of uncertainty, which is developed from information theory [6]. As the amount of available information increases, the uncertainty decreases, resulting in a decrease in entropy. Conversely, when there is less information and a higher uncertainty, this entropy increases [7].

Firstly, as Eugene Fama argued, when uncertainty arises in financial markets, it is challenging for participants to assess and respond to information accurately, resulting in price volatility [8,9]. The uncertainty arising from market volatility is closely related to fluctuations in unpredictable asset prices, highlighting the dynamic and uncertain nature of price movements, which can significantly impact investment decisions and the overall market sentiment. To measure the market uncertainty related to price volatility, various variants of entropy have been proposed based on information entropy. Claudiu Vinte introduced the approach of cross-sectional intrinsic entropy to estimate the uncertainty in stock markets [10]. This approach takes into account the trading volume and price movements of various assets, allowing for a more holistic understanding of market dynamics. As the understanding of market volatility gets deeper, some researchers have recognized its transmission effect, which can give rise to various forms of market uncertainty within the same classification. Thomas Dimpfl employed transfer entropy to quantitatively assess the transmission of volatility between different financial markets [11]. Understanding this transmission of volatility can be crucial for investors and policymakers in making informed decisions and conducting effective risk management.

Furthermore, uncertainty in portfolio selection is related to investors’ asset allocation. Investors aim to achieve objectives through the rational allocation of different types of assets. However, there are randomness and fuzziness factors in markets that prevent investors from fully predicting the returns and values of assets, which leads to uncertainty in portfolio selection [12,13]. Philippatos and Wilson were among the first to apply the concept of entropy to portfolio selection [14]. They proposed a mean entropy approach to measure the uncertainty in the asset allocation process. Their pioneering research shed light on the importance of considering uncertainty in portfolio management, leading to a paradigm shift in how investors make decisions about asset allocation. Building upon their work, more generalized forms of entropy, such as incremental entropy, were formulated. Compared to the traditional portfolio selection theory, the theory based on incremental entropy emphasizes that there is an optimal portfolio for a given probability of return [15]. Xu et al. introduced the concept of hybrid entropy and utilized it to measure the asset risk caused by both randomness and fuzziness [16]. Using information entropy to measure the level of uncertainty in portfolio selection can effectively assist investors in evaluating and optimizing their asset allocation strategies.

Finally, uncertainty in the option-pricing process is related to the impact of uncertain factors such as underlying asset price volatility and interest rates. Options are financial instruments whose value and returns depend on the price movements of the underlying assets and other factors. Les Gulko introduced the entropy pricing theory (EPT), which can provide valuation results similar to the Sharpe–Lintner capital asset-pricing model and the Black–Scholes formula [17]. His research was also extended to stock option pricing [18] and bond option pricing [19], using the EPT to measure collective market uncertainty. The EPT model demonstrates simplicity and user-friendliness, aligning with the principles of the Efficient Market Hypothesis [5]. Based on the previous analysis, it is evident that information entropy is a comprehensive tool for measuring the uncertainty in financial markets. In comparison to traditional tools, it provides a better reflection of these uncertainties that exist in financial markets.

Although much attention has been paid to the uncertainty in financial markets, it is worth noting that there are other forms of uncertainties that exist across various markets. Information asymmetry is an important factor that leads to quality uncertainty [20,21], which may lead to issues such as adverse selection [22,23]. Additionally, there is economic policy uncertainty (EPU) present in the market, referring to the impact of exogenous shocks related to economic policies that introduce unpredictability and uncertainty into the market [24]. Lots of empirical studies have shown that EPU shocks can lead to stock market turbulence [25,26]. These uncertainties are not limited to financial markets, and they can actually occur in all kinds of markets.

All the uncertainties mentioned above are important and have indeed been extensively studied in the literature. However, it should be noted that there is another significant type of uncertainty that has not received sufficient attention. This particular form of uncertainty stems from the mismatch between the quantities of desired exchanges in market transactions. In a market where buyers and sellers engage in trading activities, an equilibrium is achieved when the quantity supplied equals the quantity demanded. However, in reality, markets often operate in a disequilibrium state, where the quantities supplied and demanded are not equal with each other. This condition implies that traders may face uncertainty in their transactions. In this paper, we focus on this specific type of uncertainty and aim to put forward a metric to measure and analyze it.

Based on the foregoing analyses, the current applications of entropy in measuring the uncertainty caused by incomplete information, as well as its utilization in characterizing asset portfolios and risk assessment in financial markets, does not provide a comprehensive understanding of the mechanisms of market operation. In particular, there is no equivalent concept of entropy to express the uncertainty of participants’ trading in the market. Thus, we come up with the concept of transaction entropy to represent the uncertainty of market trade. We investigate how this transaction entropy changes with price. The results show that the equilibrium market has the lowest entropy. Additionally, we also compare the total entropy between centralized and decentralized markets, where the key distinction lies in the presence of a price-filtering mechanism. The result shows that the total entropy is lower in a centralized market than that in a decentralized market. This finding highlights the effectiveness of price filtering in reducing market uncertainty and emphasizes the importance of integrating a price-filtering mechanism in the trading process to ensure market transaction stability.

The contributions of this paper can be summarized as follows: (1) the proposal of a concept of “transaction entropy” to measure the level of uncertainty in the process of transactions. By introducing this concept, we are able to better understand and quantify market uncertainty, providing a new perspective for in-depth analyses of the mechanism of market function; (2) the addition of an alternative metric for market performance based on the existing framework of market function analyses. Through investigating the variation in transaction entropy with respect to price changes, we find that the state of market equilibrium not only corresponds to the highest volume of transaction and the maximum market surplus, but also the lowest entropy; (3) a comparison of the levels of total entropy between centralized and decentralized markets, revealing that the presence of a price-filtering mechanism enhances successful transactions and reduces market uncertainty; (4) a comparison of computational and simulation results in terms of various aspects, including the quantity of transactions, market surplus, transaction entropy, and the total entropy in centralized and decentralized markets, to verify the theoretical analysis; and (5) a clarification of the limitations of traditional market equilibrium analyses, while emphasizing the importance of transaction uncertainty.

The remaining sections of this paper are organized as follows. Section 2 formulates the functions of supply and demand based on the concept of willingness price. In Section 3, we analyze market performance, including transaction quantity, market surplus, and transaction entropy, using the rationing rate. Additionally, we compare the total entropy in centralized and decentralized markets and discuss policy implications based on the comparison results. Section 4 presents the simulation settings and results, demonstrating the generating process of each variable that characterizes market performance. In Section 5, we discuss the importance of market transaction uncertainty by highlighting the shortcomings of the Walrasian general equilibrium and Marshall partial equilibrium approaches. We also discuss the plausible applications of transaction uncertainty analyses in real-world scenarios. Section 6 draws the conclusions.

## 2. The Expression of Demand and Supply with Willingness Price

A partial equilibrium analysis (PEA) is a widely used tool for understanding market performance. It argues that supply and demand collectively represent two sides of traders in a market, making it simple to analyze the consequence of their interaction by tracing the equilibrium point and social welfare implications [27]. However, the PEA also needs to be improved, since it fails to clearly identify how sellers and buyers constitute supply and demand curves correspondingly. To solve this problem, Wang and Stanley introduced the concept of willingness price and formulated supply and demand functions to restate the PEA in a goods market [28]. The major advantage of this approach is that the laws of supply and demand can be derived directly, and the efficiency of market equilibrium can be strictly proved.

In this paper, we follow their approach to describe the supply and demand in a goods market. We assume that each trader is willing to make a trade of one unit of goods and has a willingness price before participating in the trade. For one seller, their willingness price is defined as the minimum price that they are willing to sell one unit of goods. On the other side, the willingness price of a buyer is defined as the maximum price that they are willing to spend for one unit of goods. Supposing that a seller with a willingness price $v^{s}$ meets a buyer with a willingness price $v^{b}$, their deal can be made only if $v^{b}\geq v^{s}$ is valid. Although we cannot identify all traders’ willingness prices in real markets, we know that they exist there and govern whether a deal can be made or not.

As all participants’ willingness prices are exogenously given, the willingness prices of sellers and buyers must have a distribution correspondingly. It is reasonable to assume that willingness prices spread over the domain of (0, +∞). This spread can be characterized by probability density functions, $f_{s}(v)$ and $f_{B}(v)$ for sellers and buyers, respectively. Supposing that the numbers of the sellers and buyers are given exogenously, denoted as $N_{S}$ and $N_{B},$ respectively, then we can use $F_{s}(v)=N_{s}\times f_{s}(v)$ and $F_{B}(v)=N_{B}\times f_{B}(v)$ to characterize such distributions. From the normalization condition, we have the integrals of $F_{s}(v)$ and $F_{B}(v)$ over the whole region of willingness prices, which are $N_{s}$ and $N_{B},$ respectively,
(1)$$\int_{0}^{\infty }F_{s}(v)dv=N_{s},$$
(2)$$\int_{0}^{\infty }F_{B}(v)dv=N_{B}.$$
For any one seller, given a market price of p, they will make their choice by comparing the willingness price and market price, that is to say, the necessary condition for the seller to sell one unit of goods can be expressed as
(3)$$p\geq v_{S}.$$
Otherwise, the seller will withdraw their offer.

Equation (3) implies that only the sellers whose willingness price is not greater than the actual market price are willing to sell their goods. Combining (1) and (3), we can obtain the supply function with a given market price $Q_{S}\left(p\right)$, which can be written as
(4)$$Q_{S}\left(p\right)=\int_{0}^{p}F_{s}\left(v\right)dv.$$
The above rationale can also be applied to derive the demand function. For a buyer, only if his willingness price $v_{B}$ is higher than or equal to the market price p, will he buy one unit of goods in a market, i.e.,
(5)$$p{\leq v}_{B}.$$
Otherwise, he will give up on his purchase. Combining (2) and (5), we can obtain the demand function with a given market price $Q_{D}\left(p\right)$ of the market, which is given by,
(6)$$Q_{D}\left(p\right)=\int_{p}^{\infty }F_{B}\left(v\right)dv.$$

As is well known, there are many factors that can affect the supply and demand in a market. From the expressions of supply and demand given by Equations (4) and (6), the implicit governing factor of supply and/or demand is the willingness prices of the market participants. Thus, we can infer that most relevant factors take their effects through the willingness prices of sellers and buyers. As a result, any change in any variable that impacts these willingness prices will have an impact on the supply and demand of the goods. In addition, the extent of a market determines the total quantities of the goods demanded and supplied, which also has an impact on the supply and demand functions.

Another important inference of supply and demand functions is that we can prove the laws of supply and demand by taking a derivative of these two formulas. The first derivatives of the supply and demand functions can be expressed, respectively, as the following,
(7)$$\frac{dQ_{S}}{dp}=F_{s}\left(p\right)>0,$$
(8)$$\frac{dQ_{D}}{dp}=-F_{B}\left(p\right)<0.$$
The results show that the relationship between the quantity supplied and the market price is positive. In other words, the higher market price, the more goods supplied in the market. On the contrary, the relationship between the quantity demanded and the market price is negative. Fewer goods are demanded as the price rises.

The interaction between supply and demand determines the equilibrium price level and quantity of transactions. Combining Equations (4) and (6), we can obtain the equilibrium price ${p=p}^{*}$.The equilibrium transaction quantity $T^{*}$ can be derived directly, which can be expressed as,
(9)$$T^{*}=\int_{0}^{p^{*}}F_{s}\left(v\right)\;dv=\int_{p^{*}}^{\infty }F_{B}\left(v\right)dv.$$
Figure 1 illustrates the supply and demand curves in a commodity market. The supply curve is upward sloping, and the demand curve is downward sloping. The cross-point of these two curves specifies the market equilibrium, which corresponds to the equilibrium quantity and market-clearing price of the market.

## 3. The Market Performance with Formulated Supply and Demand Functions

In this section, our primary focus is on evaluating various aspects of market performance using the newly formulated supply and demand functions. Specifically, we analyze three key dimensions: transaction quantity, market surplus, and market uncertainty caused by a quantity mismatch of the supply and demand in a disequilibrium market. To quantify this uncertainty, we propose the concept of transaction entropy, which is derived from information entropy.

### 3.1. The Quantity of Transactions

Supply and demand represent two parties of a goods market, and their interaction determines not only the market price, but also the quantity of transactions. In this section, we set the market price as being given exogenously, and investigate how transaction quantity is determined by supply and demand as the price varies.

The state of a market depends on the level of given price. The market is in equilibrium when the price makes the market clear. Otherwise, the market is in disequilibrium. This disequilibrium can be divided into two cases, one is shortage and the other is surplus. When the price is lower than the equilibrium level, it corresponds to a state of shortage, where there is more quantity demanded than the quantity supplied in the market. When the price is higher than the equilibrium level, it corresponds to a state of surplus, where there is more quantity supplied than the quantity demanded in the market. As shown in Figure 1, the regions of shortage and surplus are marked in yellow color and green color, respectively.

According to the short-side principle, the realized quantity of transactions is determined by the short side. The short side refers to the trading party with fewer willing exchanges, and those with more are at the long side. At equilibrium, the quantity supplied is equal to the quantity demanded. In this case, the quantity of realized transactions $T^{*}$ given by Equation (9) is equal to the quantity supplied and demanded.

In a shortage market, the quantity demanded exceeds the quantity supplied. Therefore, the quantity of realized transactions is determined by the quantity supplied. The expression of the realized quantity of transactions in a shortage market $T_{ST}\left(p\right)$ can be expressed as,
(10)$$T_{ST}\left(p\right)=\int_{0}^{p}F_{s}\left(v\right)dv\;\;p<p^{*}.$$
For a surplus market, the quantity demanded is less than the quantity supplied. In contrast, the quantity of realized transactions in a surplus market $T_{SP}\left(p\right)$ can be written as follows,
(11)$$T_{SP}\left(p\right)=\int_{p}^{\infty }F_{B}(v)dv\;\;p>p^{*}.$$
Based on the preceding analyses, the transaction quantity in the various states of a market can be given by,
(12)$$T\left(p\right)=\left\{\begin{array}{cc} \int_{0}^{p}F_{s}\left(v\right)dv & p<p^{*},\\ \int_{0}^{p^{*}}F_{s}\left(v\right)dv=\int_{p^{*}}^{\infty }F_{B}(v)dv & p=p^{*}\\ \int_{p}^{\infty }F_{B}(v)dv & p>p^{*}. \end{array}\right.,$$
Figure 2 shows the computational results of the relationship between transaction quantity and market price based on Expression (12), represented by the blue line. Obviously, the quantity of transactions increases with an increase in market price when $p<p^{*}$, and decreases when $p>p^{*}$. The quantity of transactions reaches its maximum when the market price attains its equilibrium level.

### 3.2. Market Surplus

#### 3.2.1. The Rationing Rates

According to the short-side principle, we know that all the participants at the long side are willing to make transactions, nevertheless, some of them cannot achieve their desired outcome. Thus, we define the rationing rate as the ratio of the quantity of actual transactions to the quantity of desired exchanges. The sellers’ and buyers’ rationing rates can be used in the following analysis of market surplus and transaction entropy. Their expressions ($G_{s}$ and $G_{B}$) are given as follows, respectively,
(13)$$G_{s}=\frac{T}{Q_{S}},$$
(14)$$G_{B}=\frac{T}{Q_{D}}.$$

It is obvious that $G_{s}$ and $G_{B}\;are$ in the range of [0, 1]. The quantities supplied and demanded will change with a variation in the market price. Therefore, the level of rationing rate will be altered as the market price varies. When the market price equals the equilibrium one, the rationing rates of either sellers or buyers equal one. Thus, we obtain,
(15)$$G_{s}(p^{*})=G_{B}(p^{*})=1.$$
In the shortage region, i.e., $p<p^{*},$ all sellers can fulfill their willing exchanges, where only a portion of buyers can successfully match with the sellers and achieve their desired transactions. As a result, the sellers’ rationing rate is one, while the buyers’ rationing rate would be less than 1. Thus, we obtain,
(16)$$G_{s}(p)=1,$$
(17)$$G_{B}\left(p\right)<1.$$
Meanwhile, with an increasing market price, there are more commodities supplied and less demanded. The rationing rate of sellers remains constant with the increase in price, while the rationing rate of buyers increases. We then obtain,
(18)$$\frac{dG_{s}\left(p\right)}{dp}=0,$$
(19)$$\frac{dG_{B}\left(p\right)}{dp}>0.$$
In contrast, the above rationale can also be applied to the surplus region, where $p>p^{*}$. The rationing rate of sellers is lower than 1, and the buyers’ rationing rate is one. Then, we obtain,
(20)$$G_{S}\left(p\right)<1,$$
(21)$$G_{B}(p)=1.$$
The relationship between the rationing rate and market price in a surplus market can also be derived. In this case, as the market price increases, sellers are less likely to obtain their rations, because the quantity supplied increases while the quantity demanded decreases. Meanwhile, the rationing rate of the buyers will not change. The derivatives of the rationing rates of sellers and buyers have the following properties,
(22)$$\frac{dG_{s}\left(p\right)}{dp}<0,$$
(23)$$\frac{dG_{B}\left(p\right)}{dp}=0.$$
Figure 3 depicts the dependence of these rationing rates on market price. As shown in this figure, when a market is in a shortage, the rationing rate of buyers is less than one, whereas the rationing rate of sellers is equal to one. In contrast, the rationing rate of sellers is smaller than 1, while the buyers’ rationing rate equals one when a market is in surplus. When a market is in equilibrium, the rationing rates of either the sellers or buyers are 1.

#### 3.2.2. The Formulation of Market Surplus

Market surplus, used to measure market efficiency, is another essential component of traditional market performance analyses. The surplus of one seller (buyer) can be defined as the difference between the actual (willingness) price and the willingness (actual) price. In the transactions of a goods market, only a portion of participants will be able to realize their willing exchanges, and a surplus will be generated. Therefore, it is reasonable to take rationing rates into account when formulizing the surplus of a market.

For sellers, given a market price *p*, the total realized surplus of these sellers ${(Z}_{sr})\;$in the market could be calculated as follows,
(24)$$Z_{sr}(p)=\int_{0}^{p}F_{s}(v)(p-v)G_{s}(p)dv.$$
On the other side, given a market price p, the total realized surplus of the buyers ${(Z}_{Br})\;$in the market could be given by,
(25)$$Z_{Br}(p)=\int_{p}^{\infty }F_{B}(v)(v-p)G_{B}(p)dv.$$
The total realized market surplus for a price $Z_{r}(p)\;$is the sum of them, i.e.,
(26)$$Z_{r}(p)=\int_{0}^{p}F_{s}(v)(p-v)G_{s}(p)dv+\int_{p}^{\infty }F_{B}(v)(v-p)G_{B}(p)dv.$$
Taking the first derivatives of Equation (26), the expression of the relationship between the derivation of surplus and market price can be expressed as,
(27)$$\frac{{\partial Z}_{r}\left(p\right)}{\partial p}=\int_{0}^{p}F_{s}\left(v\right)G_{s}\left(p\right)dv-\int_{p}^{\infty }F_{B}\left(v\right)G_{B}\left(p\right)dv\;+\int_{0}^{p}F_{s}(v)(p-v)\frac{{\partial G}_{s}(p)}{\partial p}dv+\int_{p}^{\infty }F_{B}(v)(v-p)\frac{{\partial G}_{B}(p)}{\partial p}dv.$$
Combining Equations (4)–(6), (13) and (14), Equation (27) can be rewritten as,
(28)$$\frac{{\partial Z}_{r}\left(p\right)}{\partial p}=\int_{0}^{p}F_{s}(v)(p-v)\frac{{\partial G}_{s}(p)}{\partial p}dv+\int_{p}^{\infty }F_{B}(v)(v-p)\frac{{\partial G}_{B}(p)}{\partial p}dv.$$
When the market is in a shortage, we can obtain the following expression by combining Equations (18), (19) and (28),
(29)$$\frac{{\partial Z}_{r}(p)}{\partial p}>0.$$
When the market is in surplus, we can obtain the following expression by combining Equations (22), (23) and (28),
(30)$$\frac{{\partial Z}_{r}(p)}{\partial p}<0.$$
Figure 4 depicts the relationship between market surplus and market price. From this figure, we can find that the market surplus increases when $p<p^{*}$ and decreases when $p>p^{*}$. When the market is at equilibrium, the market surplus attains its maximum.

### 3.3. Market Uncertainty

Except for traditional market performance, which focuses on transaction quantity and market surplus, we also consider market uncertainty as an additional dimension of market performance.

#### 3.3.1. Transaction Entropy

Information entropy is a commonly used tool for measuring the level of disorder and uncertainty, and its extension has been widely applied in the fields of economics and finance [5,29,30]. In this section, we introduce a new kind of information entropy, named transaction entropy, to characterize market uncertainty and investigate how transaction entropy changes as market price varies.

To figure out the information entropy of one participant, we need to identify the transaction procedure, which is shown in Figure 5. At first, the participant has to make sure whether they satisfy the price-filtering mechanism given by Equations (3) and (5). At this stage, there are only two filtering results for the participants: remain in or exit the market. The exiting participants refer to ones whose willingness prices does not satisfy the condition of trade in the market, while the remaining participants refer to those who satisfy the trading conditions. It is worth noting that it is possible to fail in the trade for the remaining participants. Only the traders in the short side can make a deal.

In one word, the possibility of the participant making a deal is uncertain. Therefore, we can use the information entropy proposed by Shannon to present the uncertainty of the trader’s transaction in the market [4]. The definition of information entropy for individual traders $H\left(E\right)$ can be written as,
(31)$$H\left(E\right)=-[E\;*\;lnE+\left(1-E\right)\;*\;\ln\left(1-E\right)],$$
where E is the possibility of a successful trade. It should be noted that the possibility of a successful trade in this case is the rationing rate, referred to in the former subsection. Therefore, the respective information entropy of one seller $H_{s}$ and one buyer $H_{B}$ can be given by, respectively,
(32)$$H_{s}\left(p\right)=-[G_{s}(p)\;*\;lnG_{s}(p)+\left(1-G_{s}(p)\right)\;*\;\ln\left(1-G_{s}(p)\right)],$$
(33)$$H_{B}\left(p\right)=-[G_{B}(p)\;*\;lnG_{B}(p)+\left(1-G_{B}(p)\right)\;*\;\ln\left(1-G_{B}(p)\right)]$$

We assume that a trader is willing to make an exchange with one unit of goods, so Equations (32) and (33) can also present the information entropy of their willingness exchange quantity. The willingness exchange quantities of the remaining sellers and buyers are denoted as $Q_{S}$ and $Q_{D}$. Combining the supply and demand functions given by Equations (4) and (6), we obtain the total information entropy TS as follows,
(34)$$TS=\int_{0}^{p}F_{s}\left(v\right)H_{s}\left(p\right)dv+\int_{p}^{\infty }F_{B}(p)H_{B}\left(p\right)dv.$$

From Equations (16) and (17), we find that the rationing rate of sellers $G_{s}\left(p\right)=1$ and rationing rate of buyers $G_{B}(p)<1$ when $p<p^{*}$. As a result, we can derive that $H_{s}\left(p\right)=0$ and $H_{B}\left(p\right)\neq 0$ directly from Equations (32) and (33). The total information entropy of the market equals the information entropy of buyers. When $p>p^{*}$, the rationing rate of sellers $G_{s}\left(p\right)<1$ and the rationing rate of buyers $G_{B}(p)=1$ is based on Equations (20) and (21), so $H_{B}\left(p\right)=0,H_{s}\left(p\right)\neq 0$. In this case, the total information entropy of the whole market equals the information entropy of the sellers, which can be obtained from Equation (34). The rationing rates of the sellers and buyers are equal to one when $p=p^{*}$, given by Equation (15), and the information entropy of the sellers and buyers is equal to zero. Thus, Equation (34) can be rewritten as,
(35)$$TS=\left\{\begin{array}{cc} \int_{p}^{\infty }F_{B}(v)H_{B}\left(p\right)dv & p<p^{*}\\ 0 & p=p^{*}\\ \int_{0}^{p}F_{s}\left(v\right)H_{s}\left(p\right)dv & p>p^{*}. \end{array}\right.$$

The expression indicates that the resulting information entropy contains the contributions of all the actual transactions. To eliminate the effect of the market scale on the information entropy, we define the transaction entropy generated by one transaction to measure the market performance. Then, the transaction entropy takes the following form,
(36)$$S=\frac{TS}{T}=\left\{\begin{array}{cc} \frac{H_{B}\left(p\right)}{G_{s}\left(p\right)} & p<p^{*}\\ 0 & p=p^{*}\\ \frac{H_{s}\left(p\right)}{G_{B}\left(p\right)} & p>p^{*}. \end{array}\right.$$

For the sake of simplicity, we denote that $G\left(p\right)=min\left\{G_{s},G_{B}\right\}.$ When the market price is lower than the equilibrium price, the minimum rationing rate between the sellers and buyers is that of the sellers. When the market price is higher than the equilibrium level, the minimum rationing rate is that of the buyers. Then, Equation (36) can be transformed into the following form,
(37)$$S=-\frac{\left[G\left(p\right)\;*\;lnG\left(p\right)+\left(1-G\left(p\right)\right)\;*\;\ln\left(1-G\left(p\right)\right)\right]}{G\left(p\right)}.$$
It is obvious that the level of transaction entropy S is non-negative. According to L’Hospital’s rule, the entropy tends to be positive infinity when the market price tends to positive infinity or zero. That is to say,
(38)$$\underset{p\to 0}{\lim}S=\underset{p\to 0}{\lim}\ln(\frac{1}{G(p)}-1)=+\infty ,$$
(39)$$\underset{p\to +\infty }{\lim}S=\underset{p\to +\infty }{\lim}\ln(\frac{1}{G(p)}-1)=+\infty .$$
Taking the first derivation of Equation (37), we can obtain,
(40)$$\frac{\partial S}{\partial p}=\frac{G^{\prime }\left(p\right)}{G^{2}\left(p\right)}\;*\;\ln\left(1-G\left(p\right)\right).$$
When the market price is lower than the equilibrium one, the relationship between the transaction entropy and market price is negative, which can be expressed as,
(41)$$\frac{\partial S}{\partial p}<0.$$
When the market price is greater than the equilibrium one, the transaction entropy and market price have a positive relation, which can be presented as,
(42)$$\frac{\partial S}{\partial p}>0.$$
Figure 6 presents the results of the relationship between the transaction entropy and market price. From the figure, we can see that the slope is downward when $p<p^{*}$, while it is upward in the case of $p>p^{*}.$ Moreover, the single equilibrium transaction entropy corresponds to zero when $p=p^{*}$.

#### 3.3.2. Total Entropy in Centralized and Decentralized Markets

In this section, we redirect our focus from analyzing the entropy generated by one transaction (*S*) to examining the total entropy (TS) within two distinct market structures: a centralized market and a decentralized market. The difference between these two markets is the presence of price filtering or not. A centralized market can be regarded as having transactions with price filtering, while a decentralized market has transactions without price filtering. By comparing the entropy in these two market types, we can reveal the role of price filtering in mitigating market uncertainty.

Firstly, we examine the total entropy in a centralized market. The centralized market is characterized by the presence of a central authority or intermediary that sets one order book to collect the bid–ask prices of traders, thereby facilitating all trading activities within the market [31,32]. It is worth noting that, in our previous analysis of transaction entropy, we assumed that a given market price serves as the reference condition for transactions, which is consistent with the key assumption of the centralized market. Therefore, we can conduct an analysis of the total entropy in a centralized market based on the existing results from the previous sections.

For the sake of simplicity, we make the following assumptions: (1) the number of sellers is equal to that of buyers, denoted as N; (2) the willingness prices of the sellers and buyers are in the range of [a,b], and both a and b are positive; and (3) the supply and demand functions are linear. With these assumptions, we can easily obtain $F_{S}\left(v\right)=F_{B}\left(v\right)=k$, where k is a constant variable. As for the total entropy, considering the foregoing assumptions, we can rewrite Equation (35) as follows,
(43)$$TS=\left\{\begin{array}{cc} \int_{p}^{b}F_{B}\left(v\right)H_{B}\left(p\right)dv & a<p<p^{*};\\ 0 & p=p^{*};\\ \int_{a}^{p}F_{s}\left(v\right)H_{s}\left(p\right)dv & p^{*}<p<b. \end{array}\right.$$
Additionally, we can express the supply and demand functions in the centralized market, denoted as $Q_{SC}\left(p\right)$ and $Q_{DC}\left(p\right),$ respectively, as follows:(44)$$Q_{SC}\left(p\right)=\int_{a}^{p}F_{S}\left(v\right)dv=k\left(p-a\right),$$
(45)$$Q_{DC}\left(p\right)=\int_{p}^{b}F_{B}\left(v\right)dv=k\left(b-p\right).$$
Taking the first derivations of (44) and (45), we obtain the following results,
(46)$$Q_{SC}^{\prime }\left(p\right)=F_{s}\left(p\right)=k,$$
(47)$$Q_{DC}^{\prime }\left(p\right)=-F_{B}\left(p\right)=-k.$$
By substituting Equations (13), (14), (32) and (33) into Equation (43), the expression of the total entropy can be rewritten as,
(48)$$TS=\left\{\begin{array}{cc} -\left[\begin{array}{c} Q_{SC}\left(p\right)\ln Q_{SC}\left(p\right)+\left(Q_{DC}\left(p\right)-Q_{SC}\left(p\right)\right)\ln\left(Q_{DC}\left(p\right)-Q_{SC}\left(p\right)\right)\\ -Q_{DC}\left(p\right)\ln Q_{DC}\left(p\right) \end{array}\right] & a<p<p^{*};\\ 0 & p=p^{*};\\ -\left[\begin{array}{c} Q_{DC}\left(p\right)\ln Q_{DC}\left(p\right)+\left(Q_{SC}\left(p\right)-Q_{DC}\left(p\right)\right)\ln\left(Q_{SC}\left(p\right)-Q_{DC}\left(p\right)\right)\\ -Q_{SC}\left(p\right)\ln Q_{SC}\left(p\right) \end{array}\right] & p^{*}<p<b. \end{array}\right.$$
To clarify the concavity of the total entropy, we can differentiate Equation (48) based on Equations (46) and (47). The results show that $\underset{p\to a}{lim}{TS}^{\prime }=+\infty ,\underset{p\to p*-}{lim}{TS}^{\prime }=-\infty$*,* and$\underset{p\to p*+}{lim}{TS}^{\prime }=+\infty$, $\underset{p\to b}{lim}{TS}^{\prime }=-\infty$, where ${TS}^{\prime }$ is the derivative of TS. Moreover, it can be observed that the second derivative of TS is negative, which is presented in Appendix A, indicating a concave shape. There are three price levels corresponding to the total entropy being down to zero, that is, p=a, p=b,and $p=p^{*}$.

Then, we turn our attention to an exploration of the total entropy in a decentralized market. The decentralized market operates without a centralized authority or intermediary, enabling participants to engage in direct transactions with one another [33]. The key characteristic of a decentralized market is the random matching of sellers and buyers for one period at a time, along with anonymous pairwise meetings involving bargaining [34,35]. The transaction process in the decentralized market is illustrated in Figure 7.

In this decentralized and random trading environment, every trader has an opportunity to engage in trading with one of the counterparty. A transaction will only be made if the buyer’s willingness price surpasses the seller’s willingness price; otherwise, the trade will not take place.

In order to make a comparison between the levels of total entropy in different markets, we keep the core assumptions presented in the centralized market. We suppose that the traders in the market only trade once at a time with one unit of goods in a random way. The probability of a successful transaction for a buyer with a willingness price of v′ is the ratio of the number of sellers with a willingness price lower than v′ to the total number of sellers. Similarly, the probability of a successful transaction for a seller with a willingness price of v′ is the ratio of the number of buyers with a price higher than v′ to the total number of buyers. Therefore, the respective expressions for the probability of a successful transaction for a seller (Es) and a buyer (EB) with a willingness price of v′ are as follows,
(49)Es=∫v′bFBvⅆν∫abFBvⅆν,
(50)EB=∫av′Fsvⅆν∫abFsvⅆν.
At this time, the total entropy in the decentralized market with random matching TSde is the sum of the buyers’ entropy and sellers’ entropy, which can be expressed as,
(51)TSde=∫abFSvdv * HSES+∫abFBvdv * HBEB=∫abFSv * HSvdv+∫abFBv * HBvdv
The result shows that the total entropy in the decentralized market with random matching is a constant variable, and the detailed calculations can be found in Appendix B. This constant entropy can be expressed as,
(52)$${TS}^{de}=k\left(b-a\right).$$
This result indicates that the total entropy is closely related to the market scale in this market, with the willingness prices of sellers and buyers not changing due to the assumption of traders only trading once at a time.

#### 3.3.3. Comparison of Total Entropy in Centralized and Decentralized Markets

By investigating the characteristics of centralized and decentralized markets, it is obvious that the most prominent difference between these two market structures lies in the role of price in market transactions. In a centralized market, the difference between the willingness price and market price acts as a criterion for sellers and buyers to enter the market. Conversely, in a decentralized market, there is no market price to guide the market participants, and they trade by random matching. Therefore, the decentralized market can be seen as operating without price filtering.

By comparing the different levels of total entropy in centralized $\left({TS}^{ce}\right)$ and decentralized markets $\left({TS}^{de}\right)$, we can shed light on the role of price filtering in the transaction uncertainty of a market. As discussed earlier in the analysis of the centralized market, the total entropy exhibits a symmetrical, double-humped, downward profile. Therefore, there exists two price levels at which the total entropy reaches its maximum. These price levels can be derived by solving the equation for the derivative of the total entropy with respect to price, i.e., $TS^{\prime }\left(p\right)=0$. The expressions for the resulting prices corresponding to the maximum entropy are as follows, and the detailed derivation can be found in Appendix C,
(53)$$p_{1}=\frac{\left(5-\sqrt{5}\right)b+\left(5+\sqrt{5}\right)a}{10},p_{2}=\frac{\left(5+\sqrt{5}\right)b+\left(5-\sqrt{5}\right)a}{10},$$
By substituting Equation (53) into Equation (48), we can obtain the maximum total entropy in the centralized market ${(TS}_{max}^{ce})\;$as follows,
(54)$${TS}_{max}^{ce}=\frac{k\;*\;(b-a)}{2}\;*\;ln\frac{\left(5+\sqrt{5}\right)}{\left(5-\sqrt{5}\right)}.$$
Comparing Equations (52) and (54), we can find that the total entropy of the decentralized market surpasses that of the centralized market for all prices. This result indicates that there is a higher uncertainty in transactions within a random-matching market compared to transactions with price filtering. Thus, it is evident that the filtering mechanism plays an effective role in reducing the transaction uncertainty and ensuring successful trading in the centralized market. Figure 8 illustrates the computational results of the total entropy in the centralized and decentralized markets. The double-humped curve is the total entropy in the centralized market, and the horizontal line on the top is the total entropy in the decentralized market.

Based on the computational and simulation results of the total entropy in centralized and decentralized markets, we can conclude that the price-filtering mechanism plays an effective role in reducing market uncertainty. This yields a direct suggestion for policymakers to mitigate market uncertainty, that is, to make the market price public information during the process of transactions between buyers and sellers.

However, how to form a proper market price is a key challenge for policymakers. If the willingness prices of the market participants are available, as commonly occurs in stock markets, a bid–ask mechanism can generate market prices continuously. When the willingness prices are private information, governments could set a market price to regulate markets. However, the possibility that the exogenously set market price is exactly equal to the equilibrium one is so low that market disequilibria are inevitable. As a result, transaction uncertainty during the trading process will present, i.e., the transaction entropy comes out. In this case, the traders on the “long side,” have to face transaction uncertainty in the market. As a response, they will adjust their bargaining prices during the transaction process to fulfill their willingness to trade until the market price converges to the equilibrium, where the transaction uncertainty is minimized.

In summary, in order to form a public market price when the willingness prices of participants are available, a bid–ask mechanism can work out. Oppositely, when these willingness prices are private, the market participants should be allowed to collectively form a market price by a competitive bargaining process. This self-organized process enables the market price to converge towards the equilibrium one. Although the resulting market price fluctuates over time, transaction uncertainty could be mitigated effectively by this way.

## 4. Simulation Results

Based on the computational results and theoretical analyses presented above, we develop an agent-based model in this section to simulate the interactions between buyers and sellers in a market and their exchange outcomes. This market system comprises N buyers and N sellers. By enabling them to make transactions, we can observe how some key variables in this market, including the transaction quantity, market surplus, and transaction entropy, change with market price.

At the beginning, we set N=200, and each trader is endowed with a willingness price before trading in the market. The willingness prices of these buyers and sellers are randomly generated within the range of [2,18], following a uniform distribution. To make the simulations meaningful, we set the market price in the model to vary within the range of [2,18]; otherwise, no transactions will occur. By following the change in market price, we can observe the trading behavior of all the traders and the overall market dynamics.

We first perform simulations of a centralized market. The price was set to increase gradually with an increment of 0.5 every step for the simulations, resulting in a total of 33 simulation results corresponding to market prices in the range of [2,18]. With a given market price, buyers and sellers can compare this with their own willingness prices and decide whether they participate in the potential trade or not. Following the rules given by Equations (3) and (5), only the screened participants have a chance to make transactions. According to the short-side principle, some participants may not be able to make a successful deal. The actual quantity of transactions is determined by the short side. Given the initial setup, the simulation results for how the quantity of transactions depends on the market price are plotted as red dots in Figure 2.

To estimate the possibility of successful trades for participants, we conducted 100 random transactions between screened buyers and sellers during the simulation process. Hence, the probability of successful trades for each participant could be computed as a ratio of the number of successful trades to 100 times. Subsequently, by calculating the average ratio of the successful transactions of all the screened buyers, we could obtain the buyers’ rationing rate with the given market price. Similarly, by calculating the average ratio of the successful transactions of all the screened sellers, we obtained the sellers’ rationing rate corresponding to the given market price. In Figure 3, the red and blue dots represent the simulation results of the rationing rates of the sellers and buyers, respectively. By comparing the rationing rate of the buyers with that of the sellers for each market price, we could further obtain the minimum rationing rates for all given market prices.

Moreover, each successful transaction in the trading process contributes to the market surplus from all the screened participants in once matching. To enhance the reliability of the estimation of the total market surplus, we repeated the random matching of the screened buyers and sellers 20,000 times and took the average as the value of the market surplus for each market price. All the simulation results are represented by the red dots in Figure 4.

Furthermore, for each participant who entered the market through price filtering, we obtained the probability of successful transactions for 100 times of random matching. Based on the calculations of probability for all participants, we could obtain the total entropy for the market. We then performed 200 repetitions of such a calculation of the total entropy and obtained its average value. The simulation results of the total entropy for all market prices are plotted as the blue dots in Figure 8. Then, we could obtain the transaction entropy by dividing the total entropy by the quantity of market transactions. The simulation results of the transaction entropy for all market prices are represented by the red dots in Figure 6.

For the simulation of the entropy in a decentralized market, we followed a similar process as that for obtaining the simulation results of the total entropy in a centralized market. In this kind of market, there is no price-filtering mechanism, so sellers and buyers are directly matched randomly. We first computed the possibility of successful transactions in the market for each participant and then obtained each agent’s information entropy accordingly. By summing up all the agents’ information entropy, we could obtain the total entropy in the market. We took an average of the total entropy by performing 200 simulations, which is plotted as a dash line in Figure 8. From all the figures mentioned above, we can see that the simulation results are in a high accordance with the computational ones, showing that the theoretical analyses are verified by such an alternative way.

## 5. Discussion

Market equilibrium is a fundamental concept in economic analyses, and its research involves two primary theories: the Walrasian general equilibrium theory and Marshallian partial equilibrium theory. The Walrasian general equilibrium theory assumes that there is an auctioneer who acts as an information center during the trading process. Prices are gradually adjusted in response to changes in supply and demand until equilibrium is achieved across all markets. However, the existence of the fictional Walrasian auctioneer has been criticized for its inconsistency with reality [36,37]. In contrast, the Marshallian partial equilibrium theory has been widely accepted by economists in market analyses with supply and demand curves. It focuses on individual markets and takes producers and consumers as the market participants, who are matched in a reverse rank during transactions [38,39]. This reverse rank matching refers to willingness bids to buy being typically arranged from high to low in the order book, and willingness asks to sell being arranged from low to high. This way of matching implies that the information of traders’ willingness prices is public, leading to transparent transactions and the absence of uncertainty in these transactions. As a result, the concept of transaction entropy is not applicable in this case. However, except for certain call auction markets, the willingness prices of traders are private information in most markets. Therefore, the partial equilibrium theory has limited applications.

In this paper, we argued that traders’ willingness prices are private information, and the transaction process can be depicted as a random matching of the market participants in the market. Specifically, in a centralized market, the price maker can just know who has entered the market after setting the market price, but is not aware of the willingness prices of the existing traders. Likewise, in a decentralized market, the willingness prices of traders, which guide them to make decisions, are not known by each other, whether they have successfully made a deal or not. Therefore, we can see that inherent uncertainty exists in actual transactions due to the unavailability of traders’ willingness prices. It is necessary to introduce the concept of transaction entropy to characterize this market uncertainty when willingness prices are private information.

Although our work is primarily a theoretical analysis, our findings can be extended to practical applications in various scenarios. Such applications involve many real markets, with stock markets serving as a prime example. In a stock market, the bid–ask mechanism dominates the trading and the market equilibrium can be obtained from the bid–ask prices posed by the market participants, without any transaction uncertainty.

However, stock markets often encounter situations of market disequilibrium, especially when they attain their upper limit or lower limit, leading to transaction uncertainty. The traders would have their responses to this uncertainty, which, in turn, exert significant effects on the market. In normal conditions, when market participants become aware of the presence of uncertainty, they actively adjust their bargaining prices during the bidding process to achieve market-clearing prices. Therefore, transaction uncertainty can enhance traders’ sensitivity to market conditions, facilitating more astute investment strategies and accelerating the convergence to an efficient market.

In contrast, in an extreme situation, transaction uncertainty can trigger intense responses and impose negative effects on the market. On one hand, the transaction uncertainty caused by a shortage may engender false prosperity and asset bubbles in the market. Investors, driven by dramatic uncertainty, may engage in excessive speculation, artificially inflating stock prices. However, such a prosperity bubble is unsustainable and could eventually burst, resulting in severe market downturns and financial losses for investors. On the other hand, the transaction uncertainty resulting from a market surplus can lead to market downturns and even cause market panic and crashes. The stock market circuit breakers witnessed during the COVID-19 pandemic are a spirited instance of this. When the market experiences substantial declines and its trading activities exceed the predefined thresholds, a trading halt is automatically executed, with the intention of preventing further market collapse. However, this circuit breaker can exacerbate short-term market panic, intensifying investors’ concerns about market instability and risks.

In conclusion, in order to maintain market stability and ensure the positive development of the financial system, we should consider the impacts of transaction uncertainty on markets when formulating risk mitigation measures.

## 6. Conclusions

Following the statistical approach from Wang and Stanley [28], in which the concept of willingness price was introduced to formulate supply and demand functions, as well as market surplus in a goods market, we expanded the metrics of market performance by introducing a new kind of information entropy to measure the transaction uncertainty in a disequilibrium market.

The first metric of market performance is the realized quantity of transactions. Given a market price in the centralized market, the realized quantity of transactions can be derived from the supply and demand functions. According to the short-side principle, the quantity of transactions is governed by the quantity supplied when the market is in a shortage, while when the market is in a surplus, the realized quantity is governed by the quantity demanded. When the market is at equilibrium, the quantity of transactions is determined by the cross-point of the supply and demand curves. We find that the quantity of transactions reaches its maximum at equilibrium, and it will decrease when the market price departs from the market-clearing point to a shortage or surplus.

The second metric is market surplus, which is a traditional index of market efficiency. In the calculation of realized market surplus, the rationing rate is indispensable, which is defined as the ratio of the actual transaction quantity to the desired one. Sellers and buyers have their rationing rates, which are dependent on the market status. It can be proved that the realized market surplus is at its highest when the market is at equilibrium, since it increases in a shortage and decreases in a surplus.

We argue that transaction uncertainty is a new dimension of market performance. To measure this kind of uncertainty, we first introduced transaction entropy to reflect the level of uncertainty in individual transactions. When a market is at equilibrium, the transaction entropy is zero. Otherwise, we will have positive transaction entropy when a market is in disequilibrium. It has a decreasing trend in a shortage, but an increasing trend in a surplus. The results indicated that there is no transaction uncertainty at equilibrium, and disequilibrium leads to a higher transaction uncertainty. We then made a comparison of the total entropy in centralized and decentralized markets and found that it is lower in a centralized market than a decentralized market. This means that the price-filtering mechanism plays a key role in reducing market uncertainty.

Finally, we argue that market uncertainty is necessary in analyzing market performance, since willingness prices are private information. Traditional approaches to market equilibrium assume that information of the willingness prices of traders is available, and traders engage in reverse rank matching when they make transactions. However, these assumptions are unrealistic, and the willingness prices of traders can only guide them to choose whether to enter market or not. Once they have entered a market, they are randomly matched to trade with each other, which must incur uncertainty in transactions.

## Figures and Tables

**Figure 1 entropy-25-01140-f001:**
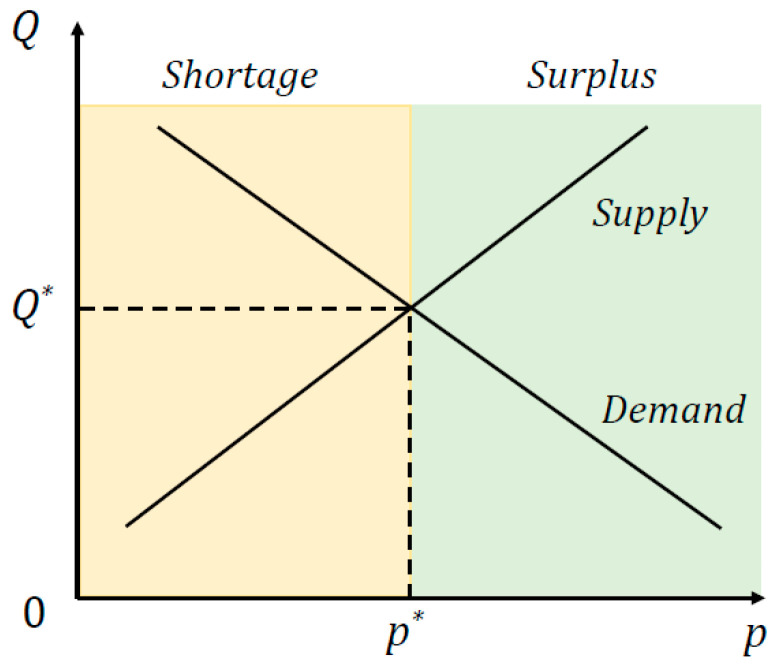
A simplified diagram of supply and demand curves in a market. The shortage region is marked in yellow color, while the surplus region is in green color.

**Figure 2 entropy-25-01140-f002:**
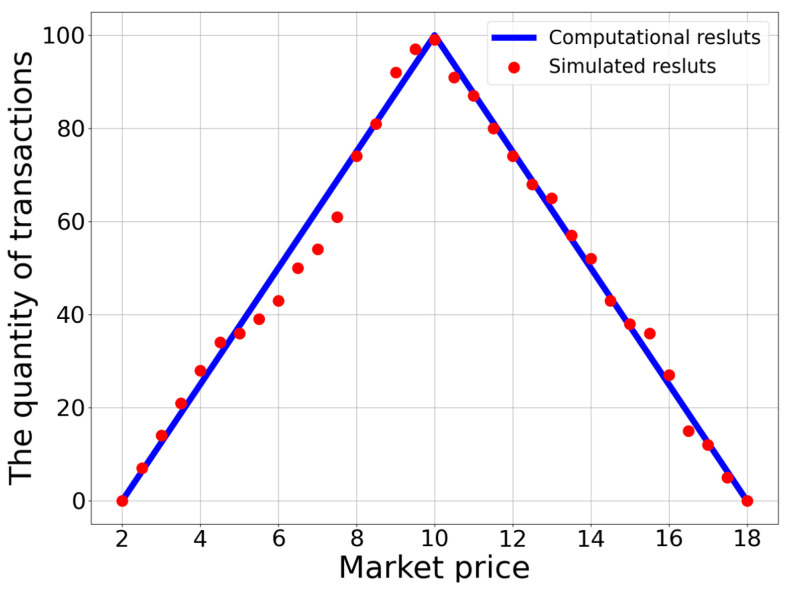
The relationship between transaction quantity and market price. The blue line represents the computational results, while red dots denote the simulation results. The participants’ willingness prices and market prices are in the range of [2,18], and the market reaches equilibrium at a price of $p^{*}=10$. For further details about the simulation settings, see the section of Simulation Results.

**Figure 3 entropy-25-01140-f003:**
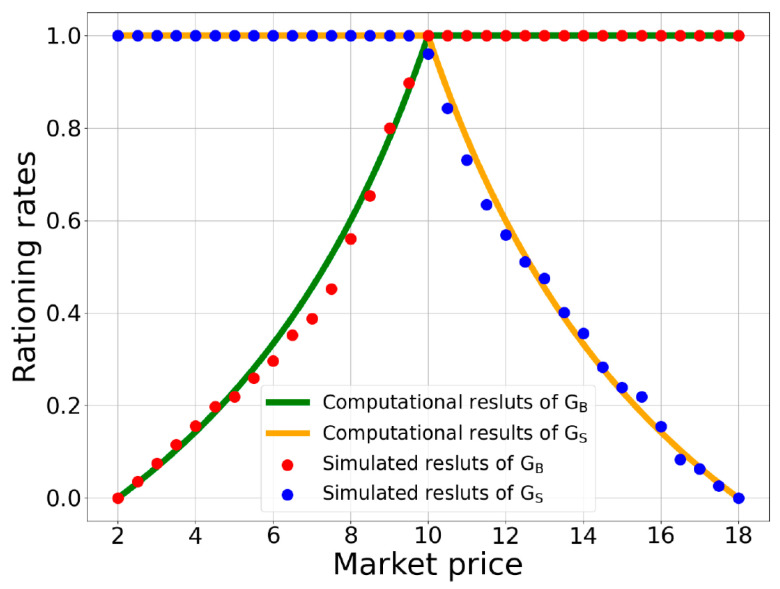
The relationship between rationing rates and market price. The green line and red dots represent the computational and simulation results of $G_{s}\left(p\right)$, respectively, while the orange line and blue dots are computational and simulation results of $G_{B}\left(p\right)$, respectively. For details about the simulation settings, see the section of Simulation Results.

**Figure 4 entropy-25-01140-f004:**
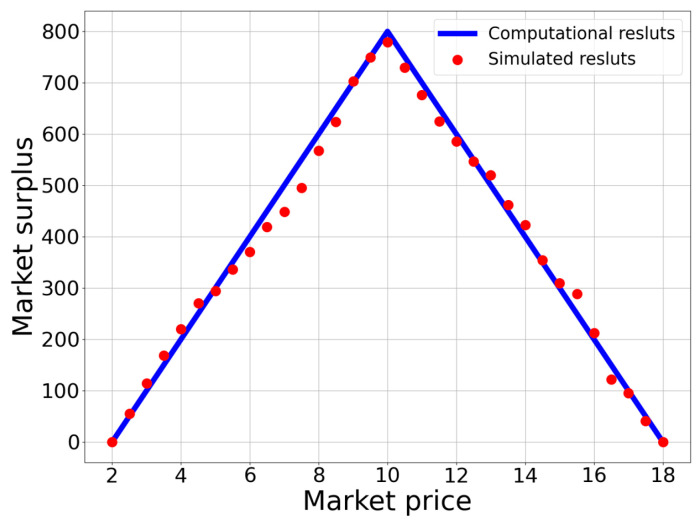
The relationship between market surplus and market price. The blue line represents the computational results, while red dots are the simulation results.

**Figure 5 entropy-25-01140-f005:**
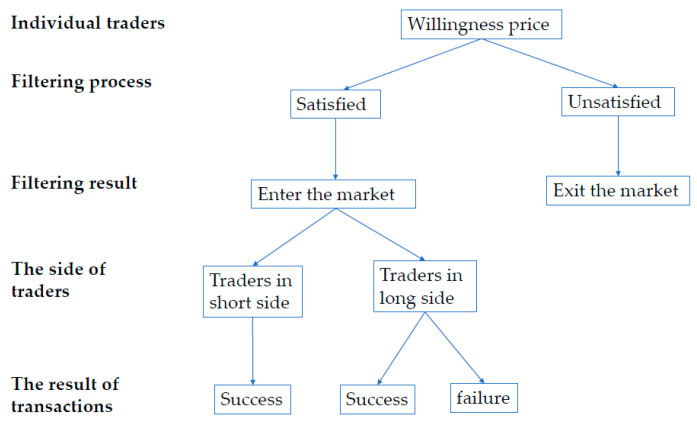
The flow chart of the transaction process for one participant’s trade in a market.

**Figure 6 entropy-25-01140-f006:**
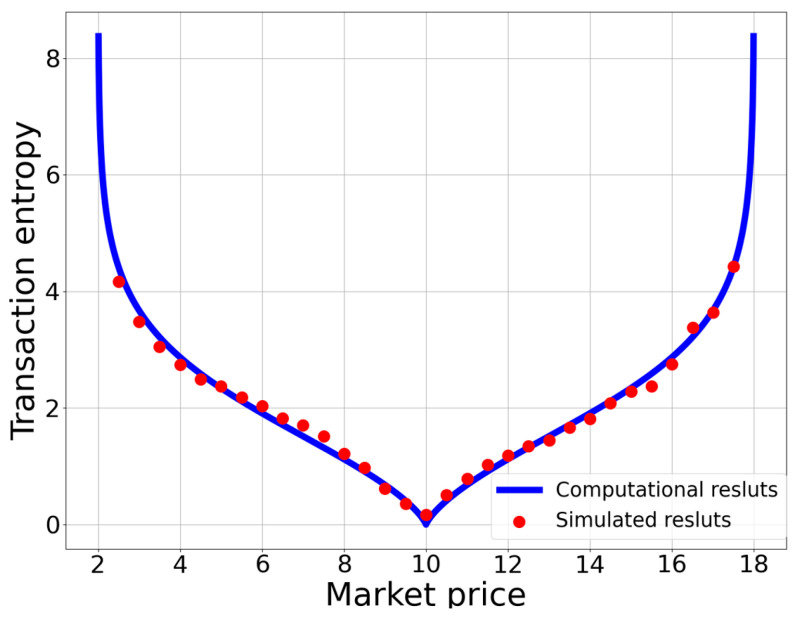
The dependence of transaction entropy on market price. The blue line represents the computational results, while the red dots represent the simulation results. When the market is in equilibrium, the transaction entropy is zero, indicating the absence of transaction uncertainty in the market.

**Figure 7 entropy-25-01140-f007:**
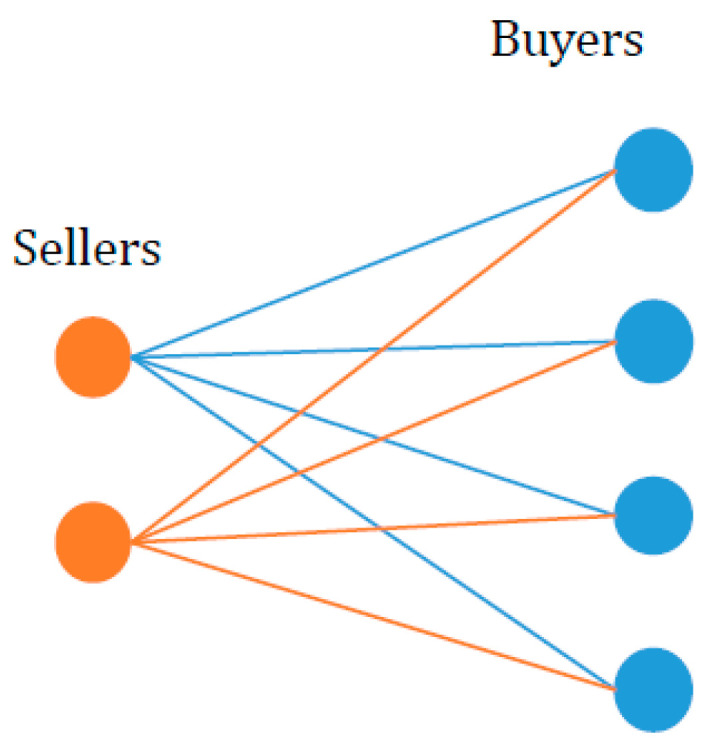
The random matching between sellers and buyers in a decentralized market. The orange circles represent sellers, and the blue circles represent buyers, each having his willingness price.

**Figure 8 entropy-25-01140-f008:**
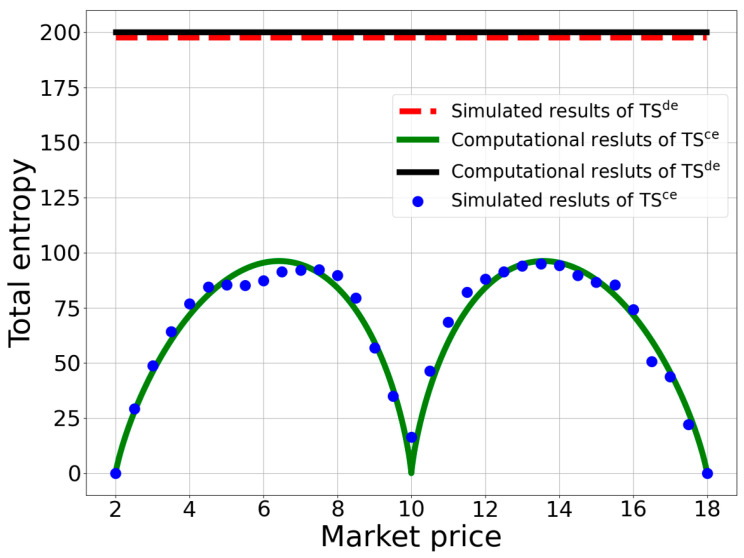
The comparison of the total entropy between centralized and decentralized markets. The green and black lines represent the computational results of centralized and decentralized markets, respectively, while the blue dots and red dash line indicate the simulation results of total entropy in centralized and decentralized markets, respectively. It is observed that the total entropy of the decentralized market is much higher than that of the centralized market at any market price.

## Data Availability

Not applicable.

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
