# Peer review of "Transaction Entropy: An Alternative Metric of Market Performance"

_entropy, 2023, doi:10.3390/e25081140_

Round 1

Reviewer 1 Report

refer to the attachments.

Null.

Author Response

Please find our responses in the attachment.

Reviewer 2 Report

Review report for the manuscript:

Transaction entropy: An alternative measurement of market performance.

The paper aims to propose the concept of transaction entropy to measure the level of uncertainty in transactions. In general, the topic could be interesting for potential readers. However, the paper is very difficult to read. Unfortunately, the research is only theoretically-oriented and it does not contain any applications for the real-data from financial markets. Therefore, the paper might be not interesting for investors and practitioners. Moreover, there are even any computer simulations that support the results and could confirm the proposed mathematical relationships and transformations. Only very simple and basic figures are presented.

Some additional major and minor comments:

·         Introduction: In the literature, the term ‘market efficiency’ means ‘market informational efficiency’ in the sense of the Efficient Market Hypothesis (EMH) proposed by prof. Fama (the Nobel Prize Winner). The Authors use this term and therefore they should refer to the EMH.

·         Introduction: As for the informational entropy, the seminal paper by Shannon (1948) is absent from the References. Moreover, the list of references is very limited although the literature concerning entropy-based applications in financial markets is vast. In this context, the literature review concerning various entropy-based applications in finance should be improved and expanded.

·         Line 149: The term ‘partial equilibrium analysis’ requires the relevant reference.

·         Lines 154 and 583: This important citation is incorrect (it should be Wang and Stanley)

·         Lines 230-231: The Authors claim that they ‘introduce a new dimension of market performance namely market uncertainty’, but the concept of market uncertainty is not new.

·         Def. (31)-(33) are exactly based on the Shannon information entropy definition, but this seminal reference is absent.

·         The theoretical deliberation should be decidedly complemented by computer simulations as this is a common practice in theoretically-oriented papers.

Author Response

(The authors gave the same response as above.)

Round 2

Reviewer 1 Report

 Accept in present form. They have addressed the queries I raised.

Reviewer 2 Report

The paper has been substantially improved.